# MM-PRM: Enhancing Multimodal Mathematical Reasoning with Scalable Step-Level Supervision

## Abstract

While Multimodal Large Language Models (MLLMs) have achieved impressive progress in vision-language understanding, they still struggle with complex multi-step reasoning. A key limitation lies in the lack of fine-grained supervision over intermediate reasoning steps. To address this, we propose MM-PRM: a unified, scalable framework for building Process Reward Models (PRMs) in multimodal settings. We first build MM-Policy-8B, a strong multimodal policy model trained on diverse mathematical reasoning data. Then, we construct MM-K12, a curated dataset of 10,000 multimodal math problems with verifiable answers, which serves as seed data. Leveraging a Monte Carlo Tree Search (MCTS)-based pipeline, we generate over 700k step-level annotations without human labeling. The resulting MM-PRM-8B is used to rerank candidate reasoning paths and achieves significant improvements across both in-domain and out-of-domain benchmarks. MM-PRM demonstrates that process supervision is a powerful tool for enhancing the logical robustness of multimodal reasoning systems. We release all our codes and data at `https://anonymous.4open.science/r/MM-PRM-F608/`.

## 1 Introduction

The rapid advancement of Large Language Models (LLMs) (Achiam et al., 2023; Bai et al., 2023a; Team et al., 2023; Touvron et al., 2023a;b; Grattafiori et al., 2024; Team, 2023; Cai et al., 2024; Guo et al., 2025) has significantly improved performance on a wide range of natural language processing tasks, including general reasoning and mathematical problem-solving. In parallel, the development of Multimodal Large Language Models (MLLMs) (Liu et al., 2023; Yao et al., 2024; Chen et al., 2024a; Wang et al., 2024c; Gao et al., 2024; Chen et al., 2024b;c; Bai et al., 2023b; Team et al., 2025) has unlocked new capabilities in vision-language understanding, showing promising results in areas such as image captioning and visual question answering (VQA). However, despite their impressive capabilities in perception and basic reasoning, MLLMs still struggle with complex multi-step reasoning tasks, particularly in mathematics. These shortcomings manifest as broken logical chains and inaccurate steps, sometimes yielding correct answers by chance, which increases false positives and reduces interpretability.

To address this issue, reward modeling (Cobbe et al., 2021; Yu et al., 2023; Uesato et al., 2022; Zang et al., 2025) has emerged as a promising paradigm. Reward models (RMs) play a central role in reinforcement learning from human feedback (RLHF) (Ouyang et al., 2022; Schulman et al., 2017; Shao et al., 2024; Lai et al., 2024; Wang et al., 2024d; Pang et al., 2024), and can also be used at inference time to select among multiple candidate responses using Test-Time Scaling (TTS) (Dong et al., 2024; Snell et al., 2024; Wang et al., 2023; Zhang et al., 2025; Feng et al., 2023; Kang et al., 2024; Ma et al., 2023; Tian et al., 2024; Zhang et al., 2024a) strategies such as Best-of-N (BoN). In general, reward models for reasoning tasks can be broadly categorized into two types: Outcome Reward Models (ORMs) and Process Reward Models (PRMs). ORMs (Cobbe et al., 2021; Yu et al., 2023) provide scalar feedback only on the final answer, overlooking the quality of the intermediate reasoning steps. This limits their ability to guide the model toward robust reasoning paths. In contrast, PRMs (Li et al., 2022; Lightman et al., 2023; Uesato et al., 2022; Wang et al., 2023; 2024e; Luo et al., 2024) offer a more fine-grained approach by evaluating each reasoning step, enabling more accurate and interpretable feedback.

Recent efforts have developed PRMs for text-only mathematical reasoning (e.g., PRM800k (Lightman et al., 2023), MathShepherd (Wang et al., 2023), OmegaPRM (Luo et al., 2024)). While these works establish valuable paradigms for process supervision, none extend to multimodal reasoning, where designing scalable data generation and stable PRM training remains an open challenge.

To address this gap, we propose **MM-PRM**, the first unified and scalable framework for building Process Reward Models (PRMs) in multimodal settings. Conceptually, MM-PRM is organized as a three-stage pipeline. In our implementation, we (i) train a multimodal policy model (**MM-Policy-8B**) on a large corpus of mathematical reasoning data to produce well-structured reasoning traces; (ii) use MCTS-based annotation with our self-collected **MM-K12** seed set of 10k verified multimodal K–12 problems to automatically generate over 700k step-level labels without human supervision; and (iii) train PRM (**MM-PRM-8B**) on the resulting data and evaluate it with Best-of-N inference across multiple benchmarks.

MM-PRM-8B demonstrates strong performance and generalization across multiple benchmarks. Although we only use process data generated from K-12-level math problems for training, MM-PRM-8B achieves remarkable results with MM-Policy-8B on benchmarks such as MathVista that improves from 62.93% to 67.60%. In addition, even though MM-PRM-8B is trained on data produced by MM-Policy-8B, it still delivers competitive results on other models like InternVL2.5-8B and InternVL2.5-78B. For instance, on the self-collected MM-K12 test set, InternVL2.5-8B improves from 27.01% to 37.80%, and on MathVerse, InternVL2.5-78B improves from 50.18% to 54.47%. These results highlight MM-PRM-8B's ability to generalize both across datasets and across model size, despite being trained on limited and fixed data sources.

Our contributions are threefold:

- A unified, scalable framework (**MM-PRM**) for building multimodal PRMs, organized as a three-stage pipeline (policy construction → process supervision generation → PRM training).

- **MM-K12**, a curated multimodal math dataset containing 10,000 seed problems and 500 test problems, all verified to contain unique, checkable answers.

- Instantiation and insights. We train **MM-PRM-8B**, which delivers strong gains across multiple benchmarks and generalizes across different models. We further provide practical guidelines for stable PRM training, showing that small learning rates and soft labels are crucial. Although evaluated on mathematical reasoning, MM-PRM is a general framework applicable to other multi-step reasoning domains such as medical decision-making or scientific QA.

## 2 RELATED WORK

### 2.1 MATHEMATICAL REASONING AND REWARD MODELING IN LLMS

Mathematical reasoning has become a focal point in evaluating the deep logical abilities of LLMs. (Cobbe et al., 2021; Hendrycks et al., 2021). Unlike general language tasks, mathematical problems demand precise multi-step reasoning and logical coherence, motivating approaches like Chain-of-Thought (CoT) prompting(Wei et al., 2022) and Self-Consistency sampling (Wang et al., 2022). Supervised fine-tuning (SFT) with structured solution datasets further reinforces reasoning performance, as demonstrated by models like Qwen (Bai et al., 2023a), InternLM (Team, 2023), and Gemini (Team et al., 2023). More recently, reinforcement learning (RL)-based optimization, exemplified by OpenAI's o1 (Jaech et al., 2024) and DeepSeek R1 (Guo et al., 2025), has emerged as a powerful strategy that surpassing SFT-only baselines.

However, despite these advances, current LLMs still frequently produce logically inconsistent reasoning steps or false-positive solutions. To address this issue, reward modeling techniques have emerged. Traditional Outcome Reward Models (ORMs)(Cobbe et al., 2021; Yu et al., 2023), which assign rewards based solely on the final answer, fail to detect flawed intermediate reasoning. Process Reward Models (PRMs)(Lightman et al., 2023; Wang et al., 2023; 2024e; Luo et al., 2024) overcome this limitation by explicitly providing step-level supervision, significantly improving logical coherence. Representative works such as PRM800K (Lightman et al., 2023), MathShepherd (Wang et al., 2023), and MiPS (Wang et al., 2024e) have demonstrated the effectiveness of PRMs in enhancing mathematical reasoning.

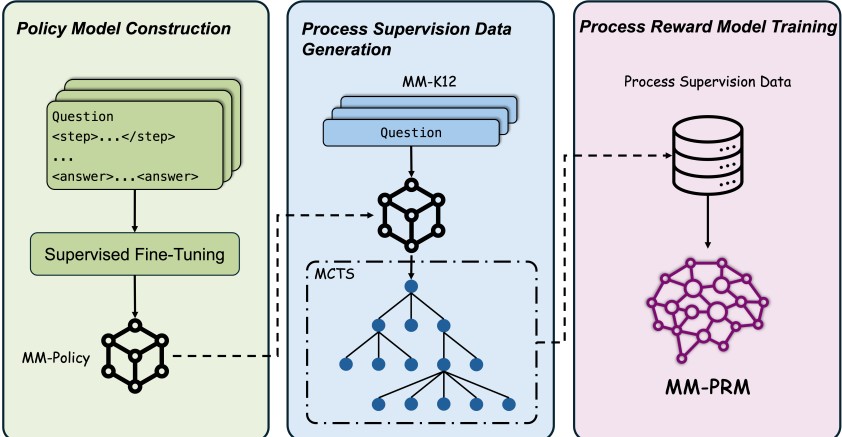

Figure 1: Overview of MM-PRM's three-stage pipeline: supervised policy model construction, MCTS-based process supervision data generation, and step-level reward model training.

## 2.2 PROCESS SUPERVISION DATA CONSTRUCTION

Training PRMs requires large-scale, high-quality step-level annotations that indicate the correctness of each intermediate reasoning step. However, most publicly available datasets focus solely on final answers, making it challenging to obtain supervision signals for reasoning quality. Existing process supervision construction methods can be broadly categorized into three classes:

**Manual Annotation.** The earliest efforts, such as PRM800K (Lightman et al., 2023), relied on human experts to label each reasoning step. While this yields high-quality annotations, the approach is costly, time-consuming, and difficult to scale—particularly for domains like mathematics where expert knowledge is required.

**Monte Carlo Estimation.** Works like MathShepherd (Wang et al., 2023) and MiPS (Wang et al., 2024e) automate supervision by sampling multiple rollouts from each reasoning prefix and estimating step correctness based on the proportion of successful completions. If continuations from a given step often lead to correct answers, that step is considered reliable. This method is simple but suffers from label instability due to high variance in sampled rollouts—especially on long or complex reasoning chains.

**Monte Carlo Tree Search (MCTS).** To address these limitations, OmegaPRM (Luo et al., 2024) introduces a structured, search-based alternative. Using a divide-and-conquer MCTS algorithm, it efficiently identifies the first error in a reasoning chain and constructs a tree of reasoning paths with dynamically updated statistics. This approach improves annotation stability, better handles deep reasoning tasks, and generates large volumes of fine-grained supervision from limited seed data without human input.

## 3 METHOD

### 3.1 OVERVIEW

We present MM-PRM, a unified and scalable framework for building PRMs in multimodal settings. Our framework addresses two critical bottlenecks in prior work: **the lack of step-level supervision** and the **inefficiency of data generation in complex reasoning tasks**. To overcome these challenges, we design a structured three-stage pipeline that enables fine-grained reward modeling without any human annotation. As shown in Figure 1, the framework comprises three interconnected stages as follows:

(1) **Policy Model Construction**, where a multimodal policy model is trained to generate high-quality reasoning traces following the CoT paradigm.

(2) **Process Supervision Data Generation**, where we use a MCTS-based engine to efficiently identify reasoning flaws and produce step-level reward labels at scale.

(3) **Process Reward Model Training**, where a PRM is trained to evaluate each reasoning step and provide dense feedback.

This end-to-end design ensures that process supervision can be generated, modeled, and applied in a fully closed loop. It significantly improves reasoning quality and robustness, especially in tasks requiring long logical chains. In Sections 3.2–3.4 below, we elaborate on these three stages.

## 3.2 POLICY MODEL CONSTRUCTION

The policy model serves as the foundation of our framework, responsible for generating candidate reasoning trajectories given multimodal math problems. These trajectories are later evaluated and labeled to form step-level supervision for training the PRM. Ensuring that the policy model produces logically coherent and structurally complete outputs is thus essential for the effectiveness of the entire system.

To train the policy model, we curated a large-scale, high-quality dataset of mathematical problems spanning a wide range of topics and difficulty levels. Our dataset integrates samples from over a dozen public math datasets, including R-CoT (Deng et al., 2024), MAVIS (Zhang et al., 2024c), MathV360K (Shi et al., 2024), NuminaMath (Li et al., 2024a), and DART-Math (Tong et al., 2024), with problems ranging from elementary school arithmetic to advanced geometry and statistics. The full list of data sources and sample counts used for training is provided in Appendix B.

Once collected, all data undergo rigorous cleaning and format standardization. Visual and textual content are paired explicitly, and reasoning traces are reformatted to follow a structured CoT schema, with each logical step clearly marked using structured tags such as `<step></step>`, and the final conclusion labeled with `<answer></answer>`. To enhance quality and clarity, we leveraged a strong instruction-tuned language model (i.e., Qwen2.5-72B-Instruct (Bai et al., 2023a)) to parse original solutions and restructure them into coherent, modular steps. After rewriting, we apply rule-based validation to discard outputs that deviate from the expected schema or contain malformed steps. This structured representation not only enhances model learnability but also lays the foundation for generating step-level reward labels in the next stage. The prompt used for solution restructuring is provided in Appendix C.

With this cleaned and annotated corpus of over 5 million examples, we fine-tuned a strong open-source multimodal model, InternVL2.5-8B (Chen et al., 2024c), using supervised learning. This ensures that the model learns to produce logically sound and well-structured outputs that conform to the CoT reasoning pattern.

The resulting model-**MM-Policy-8B**-not only delivers high-quality reasoning trajectories for downstream data annotation but also supports inference-time use cases such as BoN selection with reward-based reranking, forming the cornerstone of our scalable process supervision framework.

## 3.3 PROCESS SUPERVISION DATA GENERATION

To enable fine-grained supervision for step-level reasoning, we adopt an automated process annotation pipeline based on MCTS for efficiently identifying and labeling intermediate reasoning steps with confidence estimates.

Our process begins with self-collecting a curated dataset of 10,000 multimodal math problems from real-world named **MM-K12**—consisting of 5,000 fill-in-the-blank and 5,000 multiple-choice questions. These problems span a range of curriculum topics from elementary to high school and serve as seed instances for process supervision generation. All examples in MM-K12 are collected manually and carefully filtered to ensure that each question includes meaningful visual input and has a unique, verifiable answer, making them well-suited for structured reasoning and reward modeling. In addition, MM-K12 also provides an independent test set of 500 problems, constructed under the same criteria, which we use to evaluate in-distribution performance later. We include the distribution of mathematical domains covered by MM-K12 in Appendix D. For each problem, the policy model produces multiple candidate solutions following the CoT paradigm, and these reasoning paths form the raw material for subsequent reward annotation.

To evaluate the correctness of each intermediate step using MCTS, we generate multiple completions (rollouts) from partial prefixes and estimate the correctness of a given step based on whether its downstream completions reach the correct final answer. By applying binary search, the algorithm efficiently pinpoints the earliest step at which the reasoning begins to deviate. These supervision signals are then aggregated within a structured state-action tree, which records the Monte Carlo (MC) estimates and other statistics at each reasoning state. In our implementation, we maintain the full multimodal context—including both textual and visual components—throughout the tree construction and search process.

Importantly, our adaptation retains the efficiency of MCTS while enabling reward supervision for reasoning steps that are conditioned on complex visual stimuli. Through this pipeline, we generate over 700,000 step-level annotations from only 10k seed questions, without requiring manual labeling. The resulting dataset provides dense, high-quality process supervision aligned with real-world multimodal reasoning.

### 3.4 Process reward model training

With large-scale step-level supervision in place, we proceed to train a PRM that can assess the quality of reasoning steps given a multimodal context. The PRM is designed to serve as a fine-grained critic, assigning a reward score to each intermediate step conditioned on its preceding reasoning context that enables both test-time scaling and potential RL applications.

#### 3.4.1 Labeling strategy

A central design decision in PRM training lies in how to formulate supervision signals from the MC estimations (Luo et al., 2024; Zhang et al., 2025). Rather than adopting hard binary label (e.g., $\hat{y} = 1[\mathrm{MC}(s) > \tau]$), we use soft label, directly taking the MC scores as continuous supervision targets.

This choice is motivated by the observation that the MC score reflects more than the correctness of an intermediate step. It also encodes factors such as problem difficulty, step criticality, and distributional uncertainty in the policy model's rollouts. For instance, a reasoning step within a highly ambiguous or visually complex problem may yield lower MC scores even if the logic is fundamentally sound. In such cases, hard-thresholding may misrepresent the step's quality, introducing noise into training. By contrast, soft labels preserve the probabilistic nuance and enable smoother learning dynamics, more details will be discussed in section 5.3

Formally, for each reasoning step $x_t$ in a path $x = [x_1, x_2, \ldots, x_T]$, we assign a supervision target $\hat{y}_t = \mathrm{MC}(x_{<t}) \in [0, 1]$, where $\mathrm{MC}(x_{<t})$ denotes the estimated probability that a correct final answer can be reached from this partial path.

#### 3.4.2 Model design and training objective

To model the prediction task, we treat the PRM as a classifier operating on each reasoning step. Given a multimodal input $q$ and a generated reasoning trace $[x_1, x_2, \ldots, x_T]$, we interleave a special marker token, denoted $\sigma$, after each step, producing an input sequence of the form:

$$[q, x_1, \sigma, x_2, \sigma, \ldots, x_T, \sigma].$$

In our implementation, $\sigma$ is instantiated as the token `<prm>`. At each occurrence of $\sigma$, the model is tasked with producing a scalar confidence score indicating the likelihood that the immediately preceding step is logically correct. Formally, let $z_{\mathrm{Yes}}^{(i)}$ and $z_{\mathrm{No}}^{(i)}$ denote the unnormalized logits for binary labels "Yes" (correct) and "No" (incorrect) at the $i$-th occurrence of $\sigma$. The model's predicted probability is computed via softmax:

$$p^{(i)} = \frac{\exp(z_{\mathrm{Yes}}^{(i)})}{\exp(z_{\mathrm{Yes}}^{(i)}) + \exp(z_{\mathrm{No}}^{(i)})}.$$

The training objective is to minimize the **cross-entropy loss** between the predicted scores $p^{(i)}$ and the soft labels $\hat{y}^{(i)}$, across all scoring points:

$$\mathcal{L}_{\mathrm{PRM}} = -\sum_{i=1}^{T} \left[ \hat{y}^{(i)} \cdot \log p^{(i)} + (1 - \hat{y}^{(i)}) \cdot \log(1 - p^{(i)}) \right].$$

Table 1: Performance improvements across various benchmarks when applying the MM-PRM-8B to different models.

| Model | MM-K12 | OlympiadBench | MathVista | MathVerse | MathVision |
|---|---|---|---|---|---|
| MM-Policy-8B | 33.92 | 15.41 | 62.93 | 42.99 | 21.74 |
| *+MM-PRM-8B* | 42.80 | 24.00 | 67.60 | 46.27 | 27.11 |
| | **+8.88** | **+8.59** | **+4.67** | **+3.28** | **+5.37** |
| InternVL2.5-8B | 27.01 | 5.23 | 56.43 | 36.26 | 10.04 |
| *+MM-PRM-8B* | 37.80 | 15.33 | 63.50 | 42.56 | 19.41 |
| | **+10.79** | **+10.10** | **+7.07** | **+6.30** | **+9.37** |
| InternVL2.5-26B | 28.01 | 14.46 | 60.02 | 37.83 | 20.76 |
| *+MM-PRM-8B* | 38.00 | 24.67 | 64.50 | 44.19 | 25.63 |
| | **+9.99** | **+10.21** | **+4.25** | **+6.36** | **+4.87** |
| InternVL2.5-38B | 40.34 | 29.57 | 68.32 | 47.94 | 29.70 |
| *+MM-PRM-8B* | 52.40 | 32.67 | 71.10 | 52.61 | 32.99 |
| | **+12.06** | **+3.10** | **+2.78** | **+4.67** | **+3.29** |
| InternVL2.5-78B | 42.24 | 30.98 | 69.48 | 50.18 | 31.50 |
| *+MM-PRM-8B* | 48.80 | 34.67 | 73.20 | 54.47 | 33.26 |
| | **+6.56** | **+3.69** | **+3.72** | **+4.29** | **+1.76** |
| gpt-4o-2024-11-20 | 38.94 | 34.62 | 60.64 | - | - |
| *+MM-PRM-8B* | 49.00 | 65.90 | 42.00 | - | - |
| | **+10.06** | **+5.26** | **+7.38** | - | - |
| claude-sonnet-4 | 58.11 | 75.61 | 41.86 | - | - |
| *+MM-PRM-8B* | 65.80 | 79.40 | 45.33 | - | - |
| | **+7.69** | **+3.79** | **+3.47** | - | - |

This formulation guides the model to make fine-grained assessments of reasoning steps, assigning higher confidence to those with stronger evidence of correctness.

## 4 EXPERIMENTS

### 4.1 EXPERIMENTS SETUP

To validate the effectiveness of our proposed process reward modeling framework, we conduct a series of experiments, carefully configured to ensure fair and reproducible results.

**Policy model construction.** Our policy model(i.e., MM-Policy-8B) is initialized from the multimodal backbone InternVL2.5-8B and fine-tuned using approximately 5 million cleaned, structured math problems. The model is trained for 1 epoch with a batch size of 128 and a learning rate of 4e-5, updating only the language module while keeping the vision encoder frozen.

**Process supervision data generation.** Using MCTS-based structured rollouts, we generate approximately 747,000 step-level annotations using only 10k seed samples from MM-K12. The sampling parameters are tuned for balance between diversity and efficiency: $temperature = 1.0$, $top_k = 50$, $top_p = 0.9$, exploration coefficient $c_{\text{puct}} = 0.125$, and up to 200 search steps or 1,000 total rollouts per problem.

**Process reward model training.** We initialize the PRM from the fine-tuned policy model and train it for 1 epochs with a batch size of 512 and a learning rate of 4e-6.

### 4.2 EVALUATION STRATEGIES AND BENCHMARKS

To assess the effectiveness of MM-PRM-8B in improving reasoning quality, we adopt the BoN evaluation protocol. For each test problem, the policy model generates $N = 16$ candidate reasoning paths independently. The PRM then scores each path step-by-step, producing a sequence of floating-

point values representing the predicted quality of each intermediate step, the path with the highest score is selected as the final answer.

Since PRM outputs a vector of step-wise confidence scores for each candidate path, a crucial component of our evaluation is the *aggregation function* (Wang et al., 2024e) used to compress this vector into a scalar. We explore a diverse set of aggregation functions, including `Min`, `Average`, `Max`, `SumLogPr` (i.e., sum of log-probabilities), `SumLogOdds` (i.e., sum of log-odds), and `MeanOdds` (i.e., mean odds), each function captures different aspects of path quality. For baselines, we include both a `Random` baseline—where the final answer is randomly sampled from the same set of 16 candidates—and a strong `MajVo` (Majority Voting) baseline, which chooses the answer supported by the largest number of candidates. Formal definitions of all aggregation functions are provided in Appendix E.

We evaluate performance using answer accuracy, defined as the proportion of final selected answers that match the ground truth. This metric directly reflects MM-PRM-8B's utility in guiding the selection of correct reasoning paths. To comprehensively evaluate our model's performance and generalization, we conduct experiments on a range of multimodal math benchmarks, including MM-K12 (test set), OlympiadBench (OE_MM_maths_en_COMP) (He et al., 2024), MathVista(testmini) (Lu et al., 2023), MathVerse(testmini) (Zhang et al., 2024b), and MathVision(test) (Wang et al., 2024b). The MM-K12 test set serves as an in-distribution evaluation. For out-of-distribution assessment, we use the OE_MM_maths_en_COMP split of OlympiadBench, which contains open-ended multimodal questions from international math competitions, closely related in format to MM-K12 but significantly harder. To further test generalization, we include MathVista, which covers a wide range of visual mathematical tasks; MathVerse, which emphasizes understanding structured visual content; and MathVision, which targets abstract visual reasoning. These benchmarks provide a diverse and rigorous setting to measure both performance and generalization of our process reward modeling framework.

### 4.3 QUANTITATIVE RESULTS

We evaluate the effectiveness of MM-PRM-8B by applying it to a range of policy models and testing its impact across multiple multimodal math benchmarks.

Across all open-source models, MM-PRM-8B yields substantial performance improvements. For example, when applied to MM-Policy-8B on the MM-K12 test set, accuracy improves from 33.92% to 42.80%, and similar gains are observed with InternVL2.5-8B, where performance rises from 27.01% to 37.80%. Beyond in-domain settings, we observe that MM-PRM-8B also generalizes well to larger models and more challenging datasets. As shown in Table 1, applying MM-PRM-8B to InternVL2.5-78B improves accuracy on OlympiadBench from 30.98% to 34.67%, and on MathVerse from 50.18% to 54.47%.

To further test its generality, we also apply MM-PRM-8B to closed-source systems, including GPT-4o and Claude-Sonnet-4, and still observe consistent improvements—for instance, GPT-4o accuracy on MathVista improves by 7.38% and Claude-Sonnet-4 improves by +3.47%. Due to budget constraints, we report closed-source results only on MM-K12, OlympiadBench, and MathVista.

Despite being trained only on the small MM-K12 seed dataset and with a fixed policy model(i.e., MM-Policy-8B), MM-PRM-8B consistently enhances reasoning accuracy across diverse benchmarks and models of different sizes, training paradigms. This demonstrates the potential of scalable step-level reward modeling to improve mathematical reasoning in a model-agnostic and data-efficient manner. Detailed evaluation results across all aggregation functions and baselines are provided in Appendix F. We also provide qualitative case studies in Appendix G to illustrate how MM-PRM-8B identifies critical reasoning errors and filters low-quality trajectories.

## 5 DISCUSSION

### 5.1 CANDIDATE PATH'S IMPACT ON PRM PERFORMANCE

Since the PRM operates purely as a selector in the BoN framework, its performance is inherently bounded by the diversity and quality of candidate reasoning paths produced by the policy model.

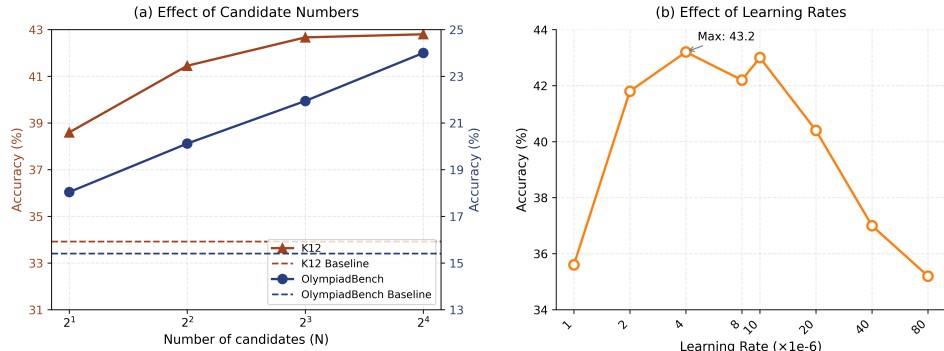

Figure 2: (a) Effect of the number of candidate reasoning paths on answer accuracy under Best-of-N inference. Increasing the number of candidates allows the PRM to select higher-quality reasoning trajectories. (b) Effect of learning rate on PRM performance. Small learning rates yield better accuracy, with performance peaking at 4e-6

In other words, PRM cannot improve a flawed generation in BoN—it can only choose among the available options. Therefore, the number of reasoning paths generated per problem directly affects its potential to identify a correct and coherent solution.

To study this effect, we vary the number of generated reasoning paths $N \in \{2, 2^2, 2^3, 2^4\}$ and measure the corresponding answer accuracy under the aggregation strategy `MeanOdds`. As shown in Figure 2(a), increasing $N$ consistently improves MM-PRM-8B's performance across both test sets. On the MM-K12 test set, accuracy improves from 38.6% at $N = 2$ to 42.8% at $N = 16$, with gains tapering off beyond $N = 8$. In contrast, on OlympiadBench, accuracy increases more steadily—from 18.4% to 24.0%—as $N$ grows. This suggests that for harder, more diverse tasks, having a larger pool of reasoning paths is critical for PRM to identify valid solutions.

## 5.2 LEARNING RATE

As noted in Lightman et al. (2023), finetuning a PRM shifts the language models' objective from generation to discrimination, making learning rate a critical factor. Smaller learning rates are often preferred to maintain stability and preserve pretrained knowledge.

We evaluate MM-PRM-8B trained under different learning rates on the MM-K12 test set using `MeanOdds` aggregator. As shown in Figure 2(b), performance peaks at 4e-6—about one-tenth the learning rate typically used in supervised fine-tuning—then drops sharply at higher values. This confirms that a moderate, conservative learning rate leads to better training, while overly large values degrade accuracy.

## 5.3 SOFT LABEL VS. HARD LABEL

Table 2: The performance comparisons of soft label vs. hard label.

| Labeling Strategy | Min | Average | Max | SumLogPr | SumLogOdds | MeanOdds |
|---|---|---|---|---|---|---|
| Soft Label | 37.4 | 43.0 | 43.4 | 42.0 | 43.2 | 42.8 |
| Hard Label | 36.8 | 34.4 | 35.6 | 36.0 | 33.8 | 37.0 |

As discussed in section 3.4.1, we adopt soft label—i.e., real-valued MC scores—as supervision for step-level reward modeling. Unlike hard label, soft label retain uncertainty and allow the model to learn more nuanced representations of reasoning quality.

To assess this design choice, we compare soft label with hard label thresholding, where steps with $MC > 0$ are treated as correct, and others as incorrect, following the protocol in Luo et al. (2024); Wang et al. (2024a).

As shown in Table 2, soft-label training consistently outperforms hard-label training across all aggregation strategies. For instance, under the `Average` aggregator, soft labels yield 43% accuracy on MM-K12 test set, compared to 34.4% with hard labels. Similar improvements are observed with `SumLogOdds` (43.2% vs. 33.8%) and `MeanOdds` (42.8% vs. 37.0%).

## 5.4 PROCESS- VS. OUTCOME-LEVEL SUPERVISION

Table 3: Comparison between MM-PRM-8B and MM-ORM-8B. MM-PRM-8B consistently outperforms MM-ORM-8B across all benchmarks.

| Model | MM-K12 | OlympiadBench | MathVista | MathVerse | MathVision |
|---|---|---|---|---|---|
| MM-PRM-8B | 42.80 | 24.00 | 67.60 | 46.27 | 27.11 |
| MM-ORM-8B | 42.40 | 20.00 | 66.60 | 45.58 | 26.18 |

To isolate the benefit of process-level supervision, we further train an alternative Outcome Reward Model (**MM-ORM-8B**) using the same data as MM-PRM-8B but discard intermediate process annotations, and instead use only the final correctness label for training. As shown in Table 3, MM-PRM-8B consistently outperforms MM-ORM-8B across all benchmarks—for example, on MathVision (27.11% vs. 26.18%) and OlympiadBench (24.00% vs. 20.00%). Gains are also consistent on high-performing datasets such as MathVista and MM-K12. These results demonstrate that process-level signals provide more informative supervision beyond final-answer correctness and leads to more reliable candidate selection during inference.

## 5.5 GENERALIZATION TO TEXT-ONLY MATHEMATICAL REASONING

Table 4: Results on the text-only subset of OlympiadBench. MM-PRM-8B consistently improves performance across different base models.

| | MM-Policy-8B | InternVL2.5-8B | InternVL2.5-38B | InternVL2.5-78B |
|---|---|---|---|---|
| Random baseline | 23.35 | 20.19 | 39.73 | 40.90 |
| +MM-PRM-8B | 29.23 | 23.89 | 41.25 | 42.43 |
| | **+5.88** | **+3.70** | **+1.53** | **+1.53** |

To evaluate whether MM-PRM's benefit extends beyond multimodal settings, we additionally test on the `OE_TO_maths_en_COMP` split of OlympiadBench, which consists of pure-text math problems drawn from olympic-level competition domains such as Algebra and Combinatorics. Compared to commonly used text-only benchmarks like MATH500, this subset focuses on challenging, competition-grade reasoning and serves as a strong testbed for evaluating general mathematical reasoning.

As shown in Table 4, MM-PRM-8B still brings consistent improvements across different policy models, e.g., +5.88% on MM-Policy-8B and +3.70% on InternVL2.5-8B. These results indicate that MM-PRM is modality-agnostic and that its benefits are not limited to multimodal inputs, but extend to mathematical reasoning more broadly, including harder symbolic domains.

## 6 CONCLUSION

In this work, we introduced MM-PRM, the first unified and scalable framework for building process reward models in multimodal settings. Our three-stage pipeline—policy construction, automatic process supervision generation, and PRM training—produces over 700k step-level annotations without human labels and yields MM-PRM-8B, which consistently improves reasoning accuracy across diverse benchmarks and models. Beyond multimodal math, MM-PRM is a general solution for step-level supervision and can be readily applied to other domains requiring structured reasoning, such as medical decision-making and scientific QA, providing a solution for building more interpretable and reliable multimodal reasoning systems.

## 7 ETHICS STATEMENT

This work complies with the ICLR Code of Ethics. No human or animal subjects were involved in this study. All datasets used, including MM-K12, were obtained in accordance with applicable usage guidelines, ensuring that privacy was not compromised. We have carefully mitigated potential sources of bias and discriminatory outcomes throughout the research process. No personally identifiable information was utilized, and no experiments were conducted that could raise privacy or security concerns. We remain committed to transparency, integrity, and ethical research practices.

## 8 REPRODUCIBILITY STATEMENT

We have taken extensive measures to ensure the reproducibility of our results. All code and datasets are publicly accessible through an anonymous repository (`https://anonymous.4open.science/r/MM-PRM-F608/`), enabling straightforward reproduction and verification. This paper provides detailed descriptions of the experimental setup, including training procedures and model configurations. Comprehensive records of experimental runs are also included in the repository to further support reproducibility. In addition, the MMK-12 dataset is fully open, and both the Qwen and InternVL model series employed in our work are open-source, ensuring consistent and verifiable evaluation results. We believe that these efforts collectively allow other researchers to reliably reproduce our work and build upon it to advance the field.

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

## A  THE USE OF LARGE LANGUAGE MODELS (LLMS)

LLMs were employed during the writing of this paper to polish the text and correct grammatical errors. The prompt used was: "Please detect and correct any grammatical errors in the following text, and polish it to enhance its academic expression. <text>"

## B  POLICY MODEL TRAINING DATA

Table 5 summarizes the datasets and sample counts used to train the policy model.

Table 5: Data sources and sample counts used for policy model training.

| Dataset | Sample Count |
|---|---|
| R-CoT (Deng et al., 2024) | 201,781 |
| MAVIS (Zhang et al., 2024c) | 29,551 |
| multimath-300k (Peng et al., 2024) | 224,747 |
| Multimodal ArXiv (Li et al., 2024b) | 99,772 |
| MathV360K (Shi et al., 2024) | 338,721 |
| Math-PUMA (Zhuang et al., 2025) | 1,111,939 |
| Multi-modal-Self-instruct (Zhang et al., 2024d) | 64,807 |
| MMPR (Wang et al., 2024c) | 1,055,796 |
| ShareGPT-4o (Cui et al., 2024) | 57,289 |
| table-vqa (Tom Agonnoude, 2024) | 80,054 |
| MathQA (Amini et al., 2019) | 27,923 |
| GSM8K (Cobbe et al., 2021) | 7,413 |
| DART-Math (Tong et al., 2024) | 574,305 |
| math-gpt-4o-200k (PawanKrd, 2024) | 196,032 |
| NuminaMath-CoT (Li et al., 2024a) | 807,883 |
| MathInstruct (Yue et al., 2023) | 257,755 |
| **Total** | **5,135,768** |

## C  PROMPT TEMPLATE FOR DATA CLEANING

The prompt template used for data cleaning is shown below. Before generating logical steps and the final answer, the language model is first instructed to explicitly identify the key knowledge points involved in the problem, along with brief explanations. This two-stage prompting strategy enhances the model's understanding of the question prior to solution restructuring.

```
Using the information provided, identify and summarize the
key  knowledge points required to solve the problem and
rewrite the original answer with a detailed reasoning process
based on the input answer.
Provide a clear explanation of each knowledge point by
stating its name followed by a colon (:), and then
presenting the detailed  Explanation on the same line.
When explaining the knowledge point, DO NOT reference or
describe  the original question, answer, or answer details.
Focus solely on explaining the knowledge points.
Besides, rewrite the original answer to include detailed
reasoning and answer. Do not change the final answer and
always refer to the input answer.

Output Format:
In order to answer this question, we first need to have the
```

```
following prior knowledge:
{{Substitute with name of knowledge point 1}}: {{Substitute
with Explanation of knowledge point 1}},
{{Substitute with name of knowledge point 2}}: {{Substitute
with Explanation of knowledge point 2}},
...
We answer this based on prior knowledge as follows:
Solution: Refined answer with detailed reasoning. Use Step 1,
Step 2 to divide the steps. Remember do not change the final
answer and always refer to the input answer.
Answer: The Final Answer is {{Substitute with final answer}}.

Input Information:

Question: {question}
-------------------------------
Answer: {answer}
-------------------------------
```

## D  TOPICAL DISTRIBUTION OF MM-K12 DATASET

MM-K12 covers a diverse set of mathematical domains, including Functions, Geometry, Probability & Statistics, and Algorithms & Flowcharts. This distribution ensures that the dataset spans a wide range of reasoning skills from symbolic manipulation to spatial understanding. The detailed topical breakdown is summarized in Table 6

Table 6: Topical distribution of MM-K12 dataset.

| Topic | Count | Percentage |
|---|---|---|
| Functions | 1149 | 10.2% |
| Geometry | 5976 | 62.9% |
| Statistics and Probability | 1093 | 11.1% |
| Algorithms and Flowcharts | 981 | 10.2% |
| Other | 801 | 5.6% |

## E  AGGREGATION FUNCTION DEFINITIONS

In the BoN inference setup, each reasoning path is assigned a vector of step-level scores predicted by the PRM. To compare and rank these paths, we apply an aggregation function to compress each score vector into a single scalar value, following the design considerations discussed in (Wang et al., 2024e). The following aggregation strategies are used in our experiments:

- Min: Computes the minimum of all step scores:

$$\text{score}(r_j) = \min\{p_1^{(j)}, p_2^{(j)}, \ldots, p_{T_j}^{(j)}\}$$

- Max: Computes the maximum of all step scores:

$$\text{score}(r_j) = \max\{p_1^{(j)}, p_2^{(j)}, \ldots, p_{T_j}^{(j)}\}$$

- Average: Computes the arithmetic mean of all step scores:

$$\text{score}(r_j) = \frac{1}{T_j} \sum_{i=1}^{T_j} p_i^{(j)}$$

- `SumLogPr`: Computes the sum of log-probabilities of all step scores:

$$\text{score}(r_j) = \sum_{i=1}^{T_j} \log p_i^{(j)} = \log \prod_{i=1}^{T_j} p_i^{(j)}$$

- `SumLogOdds`: Computes the sum of log-odds of all step scores:

$$\text{score}(r_j) = \sum_{i=1}^{T_j} \log \frac{p_i^{(j)}}{1 - p_i^{(j)}}$$

- `MeanOdds`: Computes the mean of odds-transformed step scores:

$$\text{score}(r_j) = \frac{1}{T_j} \sum_{i=1}^{T_j} \frac{p_i^{(j)}}{1 - p_i^{(j)}}$$

Here, $r_j$ denotes the $j$-th reasoning path consisting of $T_j$ steps, and $p_i^{(j)} \in (0,1)$ is the predicted correctness probability of the $i$-th step in that path.

## F  FULL EVALUATION RESULTS

### F.1  MM-POLICY-8B

Table 7: Performance comparison of various aggregators on various benchmarks for MM-Policy-8B. Top performer is in **bold**.

| Benchmark | Random | MajVo | Min | Average | Max | SumLogPr | SumLogOdds | MeanOdds |
|---|---|---|---|---|---|---|---|---|
| MM-K12 | 33.9 | 38.8 | 37.4 | 43.0 | **43.4** | 42.0 | 43.2 | 42.8 |
| OlympiadBench | 15.4 | 16.7 | 23.3 | 20.0 | **24.7** | 22.7 | 24.0 | 24.0 |
| MathVista | 62.9 | 66.7 | 66.0 | 67.2 | **67.7** | 66.4 | **67.7** | 67.6 |
| MathVerse | 43.0 | 45.6 | 46.0 | 48.0 | 46.1 | 47.3 | **48.0** | 46.3 |
| MathVision | 21.7 | 25.8 | 25.8 | 26.7 | 26.9 | 25.4 | 26.3 | **27.1** |

### F.2  INTERNVL-8B

Table 8: Performance comparison of various aggregators on various benchmarks for InternVL-8B. Top performer is in **bold**.

| Benchmark | Random | MajVo | Min | Average | Max | SumLogPr | SumLogOdds | MeanOdds |
|---|---|---|---|---|---|---|---|---|
| MM-K12 | 27.0 | 27.8 | 34.8 | 36.0 | 35.8 | 31.2 | 34.2 | **37.8** |
| OlympiadBench | 5.2 | 3.3 | 2.7 | 15.3 | **16.7** | 2.0 | 1.3 | 15.3 |
| MathVista | 56.4 | 63.0 | 63.4 | 62.8 | 62.6 | 61.0 | 62.0 | **63.5** |
| MathVerse | 36.3 | 42.1 | 41.8 | **42.8** | 41.1 | 42.0 | 42.8 | 42.6 |
| MathVision | 10.0 | 8.8 | 6.1 | 18.4 | **20.6** | 3.8 | 4.8 | 19.4 |

### F.3  INTERNVL-26B

Table 9: Performance comparison of various aggregators on various benchmarks for InternVL-26B. Top performer is in **bold**.

| Benchmark | Random | MajVo | Min | Average | Max | SumLogPr | SumLogOdds | MeanOdds |
|---|---|---|---|---|---|---|---|---|
| MM-K12 | 28.0 | 30.4 | **38.0** | 37.2 | 35.4 | 32.0 | 33.0 | **38.0** |
| OlympiadBench | 14.5 | 20.0 | 16.0 | 23.3 | 22.0 | 16.7 | 16.7 | 24.7 |
| MathVista | 60.0 | 62.0 | 63.9 | **64.6** | 64.4 | 62.8 | 64.2 | 64.5 |
| MathVerse | 37.8 | 42.0 | 42.0 | 44.2 | **44.3** | 40.1 | 41.5 | 44.2 |
| MathVision | 20.8 | 22.7 | 23.3 | 24.1 | 24.9 | 20.6 | 22.0 | **25.6** |

### F.4 INTERNVL-38B

Table 10: Performance comparison of various aggregators on various benchmarks for InternVL-38B. Top performer is in **bold**.

| Benchmark | Random | MajVo | Min | Average | Max | SumLogPr | SumLogOdds | MeanOdds |
|---|---|---|---|---|---|---|---|---|
| MM-K12 | 40.3 | 44.8 | 46.8 | 51.0 | 49.8 | 42.8 | 46.6 | **52.4** |
| OlympiadBench | 29.6 | 30.7 | 29.3 | 34.0 | **34.7** | 28.0 | 30.0 | 32.7 |
| MathVista | 68.3 | 73.2 | 70.5 | **71.1** | 70.5 | 69.2 | 70.1 | **71.1** |
| MathVerse | 47.9 | 52.5 | 50.4 | 51.9 | 51.1 | 51.1 | 52.7 | **52.6** |
| MathVision | 29.7 | 33.6 | 29.1 | 32.6 | **33.6** | 30.6 | 31.3 | 33.0 |

### F.5 INTERNVL-78B

Table 11: Performance comparison of various aggregators on various benchmarks for InternVL-78B. Top performer is in **bold**.

| Benchmark | Random | MajVo | Min | Average | Max | SumLogPr | SumLogOdds | MeanOdds |
|---|---|---|---|---|---|---|---|---|
| MM-K12 | 42.2 | 47.0 | 46.2 | 47.2 | 47.2 | 45.0 | 46.8 | **48.8** |
| OlympiadBench | 31.0 | 29.3 | 33.3 | 35.3 | **34.7** | 33.3 | **34.7** | **34.7** |
| MathVista | 69.5 | 73.0 | 71.7 | 73.4 | 72.4 | **73.9** | 73.6 | 73.2 |
| MathVerse | 50.2 | 57.9 | 53.4 | 55.0 | 54.2 | 54.3 | **55.1** | 54.5 |
| MathVision | 31.5 | 34.8 | 30.1 | 33.5 | **33.6** | 31.8 | 32.4 | 33.3 |

## G QUALITATIVE RESULTS

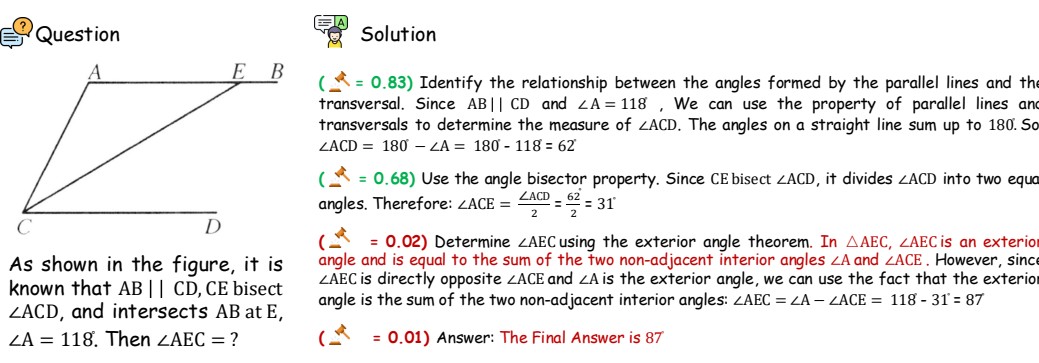

Figure 3: Qualitative example of MM-PRM-8B accurately identifying error steps in multimodal reasoning process.

To further illustrate the behavior of MM-PRM-8B, we present a qualitative example involving a geometry problem with parallel lines and an angle bisector. As shown in Figure 3, the policy model generates a four-step reasoning path, which ultimately leads to an incorrect final answer.

The PRM assigns high scores to the first two steps (0.83 and 0.68), indicating that they are logically sound. In contrast, Step-3 receives a very low score (0.02), signaling that the model has identified a significant reasoning error at this point. This flawed step leads directly to an incorrect conclusion in Step-4.

This example demonstrates that MM-PRM-8B is capable of detecting localized logical errors within a reasoning chain, such fine-grained judgment is crucial in selecting high-quality reasoning trajectories and filtering out those with subtle but critical flaws.

# H    LIMITATIONS

While our proposed framework demonstrates strong performance and generalization across multiple benchmarks, it also has several limitations: (1) Due to computational constraints, we conduct training only on the InternVL series with 8B parameters, without exploring larger models or architectures from other model families. This restricts our ability to fully assess how PRM training behavior scales with model size or generalizes across different backbones. (2) The seed data used for process supervision generation is limited in diversity, as it consists solely of K-12 math problems. As a result, the PRM may be less exposed to advanced mathematical domains or visual formats beyond the scope of standard educational settings. We leave broader model coverage and more diverse seed data construction as promising directions for future work.

