# OpenReview forum: "MM-PRM: Enhancing Multimodal Mathematical Reasoning with Scalable Step-Level Supervision"
_ICLR.cc/2026/Conference — ICLR 2026 Conference Withdrawn Submission_

### Official Review · Reviewer_gHGm · 2025-10-16

**Soundness:** 2
**Presentation:** 3
**Contribution:** 1
**Rating:** 2
**Confidence:** 4

**Summary:**

The paper proposes MM-PRM, a three-stage framework for training process reward models for multimodal mathematical reasoning: it first trains a multimodal policy model, then uses MCTS on a newly curated dataset MM-K12 to automatically generate ~700k step-level soft labels, and finally trains an 8B PRM to rerank reasoning paths at inference. Experiments show consistent accuracy gains across multiple multimodal math benchmarks and diverse base models, outperforming outcome-based reward models and demonstrating modest modality-agnostic generalization.

**Strengths:**

* Large-scale policy model and prm training data contribution.
* The paper provides clear ablations on candidates N, learning rate and labeling type that illustrates useful guidance for practical prm.

**Weaknesses:**

* The work tends toward engineering applications, lacking new insights. For example, I found that many similar works at the algorithm level or involving multimodal process supervision—such as OmegaPRM [1], ViLPRM [2],  URSAPRM [3], and VisualPRM [4]—were not included in the comparisons of MM-PRM. What are the differences in terms of data pipelines and final results compared to these approaches?

* As a reward model, why isn't MM-PRM used for online RL evaluation? To my knowledge, works such as EurusPRM[5] and DeepSeekMath [6] have already begun exploring the application of PRMs in online RL. TTS is merely a shallow validation substitute with high inference cost.

* Table 1 lacks comparisons with baselines—for example, MLLM-as-ORM, other PRMs (such as PRM400K and Math-Shepherd), or even self-consistency. I found the self-consistency comparison in Appendix F, but this should be directly included in the main table for clearer visualization.

* The improvement is marginal. For instance, in Appendix F.1, the minimum calculation and average calculation methods improve upon self-consistency by only 1.0 and 2.3 percentage points on average, respectively. Methods such as MLLM-as-ORM or MLLM-as-Generative-PRM may achieve even better TTS performance.

* Many experimental details are missing. For example, what is the inference cost during the MCTS phase? How many GPU hours were spent on rollouts? How were the parameters for rollout data generation determined? Were there empirical considerations for accuracy and diversity?

* Lack of generalization validation: the base models are primarily limited to the InternVL-series MLLMs; experiments on models such as Qwen2.5-VL, InternVL3, and Kimi-VL-MoE are necessary.

[1] Improve Mathematical Reasoning in Language Models by Automated Process Supervision. Arxiv 2406.

[2] ViLBench: A Suite for Vision-Language Process Reward Modeling. EMNLP 2025.

[3] Unlocking Multimodal Mathematical Reasoning via Process Reward Model. NeurIPS 2025.

[4] VisualPRM: An Effective Process Reward Model for Multimodal Reasoning. Arxiv 2503.

[5] Process Reinforcement Through Implicit Rewards. ICML 2025.

[6] Deepseekmath: Pushing the limits of mathematical reasoning in open language models. Arxiv 2402.

**Questions:**

* Typo in Line 291. Accuracy on mathvista seems wrong.
* Typo in Line 502. Inconsistency with MMK-12 and MM-K12.
* Cross-domain generalization: Beyond math, can MM-PRM operate on other multimodal reasoning tasks (e.g., chart QA, scientific diagrams)?
* Have you tried using MM-PRM as a critic in an RL fine-tuning loop for the policy?

---

### Official Review · Reviewer_hVgs · 2025-10-29

**Soundness:** 3
**Presentation:** 2
**Contribution:** 2
**Rating:** 4
**Confidence:** 5

**Summary:**

This paper introduces MM-PRM, a multimodal Process Reward Model that improves the performance of complex multi-step reasoning in Multimodal Large Language Models (MLLMs). By leveraging a Monte Carlo Tree Search (MCTS)-based pipeline, the authors generate over 700,000 step-level annotations without human labeling, and demonstrate substantial performance gains on both in-domain and out-of-domain benchmarks.

**Strengths:**

1. Comprehensive Data and Model Pipeline: The paper offers a detailed and well-executed approach to dataset curation (MM-K12) and the training of the MM-PRM model. The dataset, containing 10,000 multimodal math problems, is a significant contribution to the field.

2. Open-source Resources: The authors provide both the dataset and code, enabling reproducibility and further research within the community.

**Weaknesses:**

1. Clarification of the Data Cleaning Pipeline: The authors use Qwen2.5-72B-Instruct for data cleaning in the policy model construction stage. Since Qwen2.5 is not a multimodal model, could this introduce biases or incorrect visual inputs? Further discussion is needed.

2. Lack of Baseline Comparisons: The paper could benefit from comparisons against other models and approaches in the multimodal reasoning space, such as GPT or Gemini. This would provide a clearer context for evaluating MM-PRM's effectiveness.

3. Limited Ablation Studies: The paper primarily focuses on multimodal math benchmarks. It would be valuable to include ablation studies on conventional math benchmarks and other reward models to better understand the specific advantages of MM-PRM.

4. Incorporation of More Advanced Inference Techniques: The BoN evaluation method used during inference could potentially be enhanced with more sophisticated techniques. Further exploration of advanced PRM inference strategies would be beneficial.

**Questions:**

Clarify Novelty in the Pipeline: While the paper presents a solid technical approach, it would be useful to elaborate on the novelty of the MM-PRM pipeline compared to existing models.

---

### Official Review · Reviewer_SVXa · 2025-11-01

**Soundness:** 3
**Presentation:** 3
**Contribution:** 2
**Rating:** 4
**Confidence:** 2

**Summary:**

This paper presents MM-PRM, a framework for building Process Reward Models in multimodal tasks. It consists of policy construction, process supervision generation, and PRM training. The authors demonstrate how it works on math, contributing a PRM that is shown to improve test-time scaling performance. The paper also makes a dataset contribution.

**Strengths:**

+ The paper is well-written and overall easy to follow.
+ The improvement and generalization ability of MM-Policy-8B seems well-supported.
+ The analysis in Section 5 appears well-done.

**Weaknesses:**

- The central contribution, the MM-PRM framework, does not seem to be as novel as the authors claim it to be. The paper "VisualPRM: An Effective Process Reward Model for Multimodal Reasoning" has proposed something very similar.
- The 10k math problems in MM-K12 are all collected from existing benchmarks. The authors say that human verification is performed to select questions from the existing benchmarks. What are some inclusion criteria? What are characteristics of included/excluded problems? What is the filtering ratio? These details appear to be missing. If the paper wants to claim the dataset as one of its core contributions I would expect to see more originality or proof of improved quality in MM-K12 compared to the original benchmarks.
- The authors frame MM-PRM as generalizable across MM settings, but have only evaluated it on math. It is unclear how this can generalize to other domains.

**Questions:**

- The paper says "By applying binary search, the algorithm efficiently pinpoints the earliest step at which the reasoning begins to deviate." How does this work exactly? Is there a threshold of MC score below which you consider a node a deviation?

---

### Official Review · Reviewer_ADDG · 2025-11-07

**Soundness:** 2
**Presentation:** 3
**Contribution:** 2
**Rating:** 6
**Confidence:** 4

**Summary:**

This paper introduces MM-PRM, a three-stage framework that adapts process reward modeling to multimodal math reasoning. The pipeline is straightforward: a vision-language policy model is trained to produce step-by-step traces; an MCTS procedure auto-labels intermediate steps for correctness from a curated K-12 seed set (no human annotators); and a process reward model is then trained on these labels to re-rank Best-of-N solution paths at inference. In my view, the contribution is primarily a careful engineering extension of text-only PRMs to the vision-text setting, coupled with a sensible data recipe (MM-K12) and a set of practical training tips (notably small learning rates and soft step labels). Empirically, MM-PRM acts as a drop-in selector that yields consistent, non-trivial improvements on standard visual-math benchmarks and across smaller and larger backbones, without changing the base generator. The paper positions MM-PRM as a “unified, scalable” framework and suggests applicability beyond math; I find the core pipeline clear and replicable, though the evidence presented is strongest within the math domain, with claims of broader generality reading as promising but still to be demonstrated.

**Strengths:**

- Modular, reproducible design. The three-stage flow—policy training → MCTS step-labeling → PRM re-ranking—is easy to reason about, isolates responsibilities, and can be dropped into existing VLM stacks without retraining the generator end-to-end.

- Human-free process supervision at scale. Using MCTS to localize first-error steps produces dense, step-level signals from a small, curated seed—practically valuable when human annotation of chains is infeasible.

- Consistent test-time lifts as a selector. PRM re-ranking yields reliable accuracy bumps across several visual-math benchmarks and backbone sizes; it’s a low-risk, plug-in way to monetize extra sampling (Best-of-N) rather than redesigning the model.

- Actionable training guidance. Small learning rates and soft labels make PRM training more stable and performant—concrete knobs practitioners can reuse without extensive hyper-sweeps.

- Reasonable data curation. A seed set with unique, verifiable answers enables automatic labeling and trustworthy evaluation; that choice aligns well with the process-supervision objective.

- Evidence of portability. The PRM trained in one setup transfers to other model sizes/backbones with positive deltas, suggesting the scorer is not overly tied to a single policy.

**Weaknesses:**

- Incremental novelty. The contribution largely ports known text-PRM + MCTS pipelines to the multimodal setting; there’s limited algorithmic innovation beyond adding an image encoder and adapting prompts.

- Scope overreach. Claims of generality to non-math domains are not empirically supported; the approach leans on tasks with deterministic verifiers, which many target domains lack.

- Opaque compute economics. End-to-end cost (MCTS rollouts, PRM training, Best-of-N inference) is not quantified, making it hard to judge practicality vs. simpler ensembling/self-consistency.

- Selector ceiling and N-dependence. PRM cannot exceed the best sampled candidate; gains saturate with (N) and hinge on the base policy being strong/diverse enough—limits not fully characterized.

- Label quality unvalidated. No human audit or noise analysis of MCTS step labels; if labels conflate “path leads to success” with “local step correctness,” PRM may learn policy-specific shortcuts.

- Aggregator sensitivity. Multiple heuristics are explored to collapse step scores, but the choice appears tuned; without a fixed or learned aggregator, there’s risk of cherry-picking per dataset.

**Questions:**

- Why the GPT-4o MathVista drop with PRM? Please analyze failure cases and state the aggregator used for all closed-source rows.

- Compute budgets. Report wall-clock/GPU costs for (a) MCTS labeling, (b) PRM training, (c) BoN inference at N=16/32.

- Baseline gaps. Add self-consistency/majority vote and a tuned outcome-reward re-ranker as baselines to quantify PRM advantage.

- Label auditing. Provide a small human audit of MCTS step labels (precision/recall on “correct step”) and discuss mitigation if noisy.

- Fix the aggregator. Re-run with a single pre-specified aggregator (or a learned aggregator) across all datasets to remove tuning confound.

- Beyond selection. Any preliminary results on PRM-guided policy training (RL/DPO) to move gains from reranking into the generator?

- Topic imbalance. Report per-topic results on MM-K12 and, if possible, rebalanced training to test sensitivity.

---

### Note · Authors · 2026-01-05

I have read and agree with the venue's withdrawal policy on behalf of myself and my co-authors.